# Development of Improved Spectrophotometric Assays for Biocatalytic Silyl Ether Hydrolysis

**DOI:** 10.3390/biom14040492

**Published:** 2024-04-18

**Authors:** Yuqing Lu, Chisom S. Egedeuzu, Peter G. Taylor, Lu Shin Wong

**Affiliations:** 1Manchester Institute of Biotechnology, University of Manchester, Manchester M1 7DN, UK; yuqing.lu-3@postgrad.manchester.ac.uk (Y.L.); chisom.egedeuzu@manchester.ac.uk (C.S.E.); 2Department of Chemistry, University of Manchester, Manchester M13 9PL, UK; 3School of Life Health and Chemical Sciences, Open University, Milton Keynes MK7 6AA, UK; peter.taylor@open.ac.uk

**Keywords:** organosiloxane, enzyme, silicatein, hydrolase, colorimetric assay

## Abstract

Reported herein is the development of assays for the spectrophotometric quantification of biocatalytic silicon−oxygen bond hydrolysis. Central to these assays are a series of chromogenic substrates that release highly absorbing phenoxy anions upon cleavage of the sessile bond. These substrates were tested with silicatein, an enzyme from a marine sponge that is known to catalyse the hydrolysis and condensation of silyl ethers. It was found that, of the substrates tested, *tert*-butyldimethyl(2-methyl-4-nitrophenoxy)silane provided the best assay performance, as evidenced by the highest ratio of enzyme catalysed reaction rate compared with the background (uncatalysed) reaction. These substrates were also found to be suitable for detailed enzyme kinetics measurements, as demonstrated by their use to determine the Michaelis−Menten kinetic parameters for silicatein.

## 1. Introduction

Organosiloxanes (compounds presenting a C-Si-O chemical motif) are an important group of compounds that include silicone polymers [1] and protecting groups in multi-step organic synthesis [2,3]. However, their chemical synthetic manipulation typically requires harsh conditions and produces environmentally undesirable by-products. In efforts to improve the sustainability of silicon chemistry, researchers have been investigating enzymatic methods for the cleavage and formation of the Si-O bond [4,5,6,7]. Over the years, a host of hydrolases, including proteases, esterases, lipases and silicateins, have been investigated for their ability to catalyse the hydrolysis of Si-O bonds in a range of molecules [7,8,9,10,11]. Several of these enzymes have further been demonstrated to catalyse bond condensation from the corresponding silanol and alcohol though a change in the reaction conditions and hence reaction equilibrium [8,9,12].

In order to facilitate the discovery of new enzymes that can manipulate Si-O bonds and the recombinant engineering of existing enzymes, it is necessary to develop methods to detect and quantify biocatalytic Si-O bond hydrolysis. To perform such activity measurements in a convenient manner, a UV–VIS spectrophotometric assay has previously been reported using 4-nitrophenoxy silyl ethers as substrates (**1**–**3**, Figure 1) [9,10]. Here, the hydrolysis of the Si-O bond in the substrate results in the release of a *p*-nitrophenoxy anion that absorbs strongly at approximately 400–405 nm. The presence of this chromophore may also produce a visually observable yellow colour in some cases (i.e., the assay can also be colorimetric).

As an example, *tert*-butyldimethyl(4-nitrophenoxy)silane (**1**) has been used to quantify the hydrolytic activity of silicatein-α (Silα, the most common isoform of this enzyme), and subsequently to enable the determination of its Michaelis−Menten kinetic parameters [9,10]. However, it was found that this substrate generally provided a high rate of background hydrolysis, even in the absence of the enzyme under the typical assay conditions. Consequently, determining the net rate of hydrolysis (i.e., the difference between the uncatalysed and catalysed hydrolysis) tended to be less accurate at low substrate concentrations or when the enzymes exhibited low levels of activity.

Hence, the development of improved spectrophotometric assays for the quantification of Si-O bond hydrolysis is reported herein, involving the design of a set of substrates with improved stability under aqueous conditions and the tuning of the assay reaction conditions. The pro-chromogenic substrates were designed and chemically synthesised using the *tert*-butyldimethylsilyl (TBDMS) group as the silyl component. The assays were subsequently evaluated using a previously reported Silα fusion protein [10] to determine the net rates for the enzymatic reactions. The Michaelis−Menten kinetic parameters for the various substrates and this enzyme were determined as an example application.

## 2. Materials and Methods

### 2.1. Materials and Equpment

All of the solvents and reagents were of analytical grade and were purchased from either Sigma-Aldrich, Fluorochem, VWR, or Fisher Scientific. All of the buffer solutions were prepared with deionised water. Streptavidin affinity chromatography was carried out using StrepTrap HP (GE Healthcare, Amersham, UK) columns on an ÄKTA purifier chromatography system (GE Healthcare, UK). UV–VIS spectrophotometry was carried out using Synergy H1 and HT Multi-Mode Microplate Readers (BioTek Instruments, Winooski, VT, USA). Substrates **1**–**3** were synthesised according to previously reported methods [9]. The silicatein-α fusion protein production and isolation was carried out according to previously reported methods [10].

### 2.2. Hydrolysis Assays with TF-Silα-Strep

The assays were carried out according to previously published methods [9,10]. For experiments investigating the effect of pH, the purified TF-Silα-Strep was first dialysed in the desired buffer (50 mM Tris, 100 mM NaCl, at the desired pH) overnight. In addition, the UV−VIS absorbances for each substrate reaction were recorded at 414, 394, 398, and 294 nm, corresponding to the λ_max_ for substrates **4**, **5**, **6**, and **7**, respectively. The corresponding product phenolates were quantified with calibration curves constructed from known concentrations of each corresponding phenol. The initial rate (*V*_0_) was obtained by a linear fit of the data from the first 30 min of the hydrolysis reaction using previously described procedures, with all of the experiments being carried out in technical triplicates [10].

### 2.3. General Procedure for the Synthesis of Substrates ***4**–**10***

The corresponding phenol (9.98 mmol) and imidazole (19.96 mmol) were dissolved in anhydrous DMF (10 mL). *tert*-butylchlorodimethylsilane (9.98 mmol) was added to the reaction mixture and stirred at room temperature for 16 h. The reactions were all found to have reached completion after 16 h by TLC (Hexane/EtOAc, 20:1). The reaction mixture was quenched through the addition of H_2_O (10 mL) and the mixture extracted with hexane (3 × 10 mL). The organic extracts were combined, dried with MgSO_4_, filtered, and evaporated under reduced pressure. The residue was purified by silica gel chromatography to yield the desired product.

#### 2.3.1. *tert*-Butyldimethyl(2-methyl-4-nitrophenoxy)silane, **4**

The desired product was isolated as a yellow solid (1.2 g, 45%); R_f_ 0.42 (Hexane/EtOAc, 20:1); ν_max_ (liquid)/cm^−1^ 2929 (C-H), 1515 (NO_2_), 1280 (SiOAr), 1254 (C-O), 834 (Si-C); δ_H_ (400 MHz; CDCl_3_) 8.06 (d, *J =* 2.8 Hz, 1H), 7.99 (dd, *J* = 8.9 & 2.8 Hz,1H), 6.80 (d, *J* = 8.9 Hz, 1H), 2.27 (s, 3H), 1.02 (s, 9H), 0.28 (s, 6H); δ_C_ (100 MHz; CDCl_3_) 159.74 (Ar C-O), 141.29 (Ar C-N), 129.98 (Ar C-CH_3_), 126.39 (Ar C-H), 122.95 (Ar C-H), 117.68 (Ar C-H), 25.42 (*t*-butyl CH_3_), 18.11 (Si C-CH_3_), 16.73 (Ar C-CH_3_), −4.37 (Si-CH_3_); *m*/*z* (ESI^+^) 268 ([M+H]^+^, 100%); HRMS calculated for C_13_H_21_NO_3_Si [M+H]^+^: 268.1363, found: 268.1366, δ 0.9 ppm.

#### 2.3.2. *tert*-Butyldimethyl(3-methyl-4-nitrophenoxy)silane, **5**

The desired product was isolated as a pale yellow oil (1.28 g, 49%); R_f_ 0.46 (Hexane/EtOAc, 20:1); ν_max_ (liquid)/cm^−1^ 2955 (C-H), 1579 and 1339 (NO_2_), 1281 (SiOAr), 1251 (C-O), 824 (Si-C); δ_H_ (400 MHz; CDCl_3_) 8.02 (d, *J =* 9.5 Hz, 1H), 6.73 (m, 2H), 2.59 (s, 3H), 0.99 (s, 9H), 0.25 (s, 6H); δ_C_ (100 MHz; CDCl_3_) 160.46 (Ar C-O), 142.21 (Ar C-N), 137.34 (Ar C-CH_3_), 127.83 (Ar C-H), 124.02 (Ar C-H), 118.35 (Ar C-H), 25.99 (*t*-butyl CH_3_), 21.87 (Ar C-CH_3_), 18.69 (Si C-CH_3_), −3.91 (Si-CH_3_); *m*/*z* (ESI^+^) 267 ([M+H]^+^, 100%); HRMS calculated for C_13_H_21_NO_3_Si [M+H]^+^: 268.1363, found: 268.1366, δ 0.9 ppm.

#### 2.3.3. *tert*-Butyldimethyl(3-methoxy-4-nitrophenoxy)silane, **6**

The desired product was isolated as a pale yellow solid (1.3 g, 46%); R_f_ 0.45 (Hexane/EtOAc, 20:1); ν_max_ (liquid)/cm^−1^ 2930 (C-H), 1581 (NO_2_), 1281 (SiOAr), 1251 (C-O), 824 (Si-C); δ_H_ (400 MHz; CDCl_3_) 7.91 (d, *J =* 8.9 Hz, 1H), 6.48-6.42 (m, *2*H), 3.91 (s, 3H), 0.99 (s, 9H), 0.25 (s, 6H); δ_C_ (100 MHz; CDCl_3_) 162.13 (Ar C-O), 156.03 (Ar C-OCH_3_), 133.82 (Ar C-N), 128.56 (Ar C-H), 111.95 (Ar C-H), 105.43 (Ar C-H), 56.84 (Ar C-OCH_3_), 25.96 (*t*-butyl Me), 18.69 (Si C-CH_3_), −3.92 (Si-CH_3_); *m*/*z* (ESI^+^) 284 ([M+H]^+^, 100%); HRMS calculated for C_13_H_21_NO_4_Si [M+H]^+^: 284.1313, found: 284.1316, δ 1.2 ppm.

#### 2.3.4. *tert*-Butyldimethyl(4-cyanophenoxy)silane, **7**

The desired product was given as a white solid (0.8 g, 67%); R_f_ 0.46 (Hexane/EtOAc, 9:1); ν_max_ (soild)/cm^−1^ 2930 (C-H), 2260 (C-N), 1281 (SiOAr), 1251 (C-O), 824 (Si-C); δ_H_ (400 MHz; CDCl_3_) 7.54 (d, *J =* 8.6 Hz, 2H), 6.88 (d, *J* =8.6 Hz, 2H), 0.98 (s, 9H), 0.23 (s, 6H); δ_C_ (100 MHz; CDCl_3_) 159.70 (Ar C-O), 134.01 (Ar C-H), 120.87 (Ar C-H), 119.23 (C-N), 104.64 (Ar C-CN), 25.53 (Si C-CH_3_), 18.23 (Si C-CH_3_), −4.40 (Si-CH_3_); *m*/*z* (ESI^+^) 234 ([M+H]^+^, 100%). Data are consistent with the literature [13].

#### 2.3.5. *tert*-Butyldimethyl(2-methyl-4-cyanophenoxy)silane, **8**

The desired product was provided as a colourless oil (115 mg, 40%); ν_max_ (liquid)/cm^−1^ 2930 (C-H), 2260 (C-N), 1281 (SiOAr), 1251 (C-O), 824 (Si-C); δ_H_ (400 MHz; CDCl_3_) 7.43 (d, *J* = 2.1 Hz, 1H), 7.37 (dd, *J* = 8.3 and 2.1 Hz, 1H), 6.79 (d, *J* = 8.3 Hz, 1H) 2.21 (s, 3H), 1.01 (s, 9H), 0.25 (s, 6H). δ_C_ (100 MHz; CDCl_3_) 157.82 (Ar C-O), 134.56 (Ar C-H), 131.19 (Ar C-H), 130.39 (Ar C-H), 119,27 (Ar C-H), 118.57 (C-N), 103.95 (Ar C-CN), 25.43 (Si C-CH_3_), 18.08 (Si C-CH_3_), 16.46, (Ar C-CH_3_), −4.39 (Si-CH_3_); *m*/*z* (ESI^+^) 248 ([M+H]^+^, 100%); HRMS calculated for C_14_H_21_NOSi [M+H]^+^: 248.1465, found: 248.1456, δ 3.5 ppm.

#### 2.3.6. *tert*-Butyldimethyl(3-methyl-4-cyanophenoxy)silane, **9**

The desired product was provided as a colourless oil (86.76 mg, 23%); ν_max_ (liquid)/cm^−1^ 2938 (C-H), 2260 (C-N), 1281 (SiOAr), 1251 (C-O), 824 (Si-C); δ_H_ (400 MHz; CDCl_3_) 7.46 (d, *J* = 8.4 Hz, 1H), 6.74 (d, *J* = 2.3 Hz, 1H), 6.70 (dd, *J* = 8.4 & 2.3 Hz 1H), 2.48 (s, 3H), 0.98 (s, 9H), 0.22 (s, 6H). δ_C_ (100 MHz; CDCl_3_) 159.47 (Ar C-O), 144.12 (Ar C-H), 134.16 (Ar C-H), 121.87 (Ar C-H), 118.58 (Ar C-H), 118.09 (C-N), 105.20 (Ar C-CN), 25.54 (Si C-CH_3_), 20.56 (Ar C-CH_3_), 18.21 (Si C-CH_3_), −4.37 (Si-CH_3_); *m*/*z* (ESI^+^) 248 ([M+H]^+^, 100%); HRMS calculated for C_14_H_21_NOSi [M+H]^+^: 248.1465, found: 248.1456, δ 3.5 ppm.

#### 2.3.7. (*E*)-1-(4-((*tert*-butyldimethylsilyl)oxy)phenyl)-2-(4-nitrophenyl)diazene, **10**

The desired product was provided as a red solid (178 mg, 60%); R_f_ 0.63 (Hexane/EtOAc, 20:1); ν_max_ (solid)/cm^−1^ 2928 (C-H), 1492 (N=N), 1339 (NO_2_), 1258 (SiOAr), 854 (C-N), 781 (Si-C); δ_H_ (400 MHz; CDCl_3_) 8.35 (d, *J* = 12 Hz, 2H), 7.97 (d, *J* = 8 Hz, 2H), 7.90 (d, *J* = 8 Hz, 2H), 6.97 (d, *J =* 2 Hz, 2H), 1.01 (s, 9H), 0.27 (s, 6H); δ_C_ (100 MHz; CDCl_3_) 160.52 (Ar C-O), 156.49 (Ar C-N), 148.75 (Ar C-N), 147.76 (Ar C-N), 123.6 (Ar C-H), 121.15 (Ar C-H), 26.07 (Si-C-CH_3_), 18.76 (Si-C-CH_3_), −3.86 (Si-CH_3_); *m*/*z* (ESI^+^) 358 ([M+H]^+^, 100%); HRMS calculated for C_18_H_23_N_3_O_3_Si [M+H]^+^: 358.1581, found: 358.1584, δ 0.7 ppm.

## 3. Results and Discussion

### 3.1. Design and Synthesis of Substrates

Based on the previously reported substrate **1**, a series of substrates were designed that were intended to increase the hinderance of the corresponding Si-O bond towards hydrolytic attack (steric effects), decrease the nucleofugality of the phenolate leaving group (electronic effects), or a combination of both. These included **4**–**6**, where various electron-donating substitutions were incorporated around the 4-nitrophenyl ring. The analogous 4-cyanophenoxy silyl ethers **7**–**9** were also prepared, as the cyanophenoxy moiety would be a poorer leaving group [14,15], yet still be detectable by UV–VIS spectroscopy [16,17]. In addition, substrate **10**, incorporating the azo dye 4-(4-nitrophenylazo)phenol as the chromophore, was also investigated. This dye with a *tert*-butyldiphenylsilyl group had previously been reported as a colorimetric probe for fluoride ions [18]. In this case cleavage of the Si-O bond releases the dye anion, which exhibits an absorbance at ~474 nm (in the red region of the visible spectrum) that would be far removed from the absorbance of the substrate or any other potentially interfering molecules in an assay. In all cases, these substrates were synthesised by the silylation of the corresponding phenols and silyl chloride under basic conditions, with moderate yields between 45–70%.

### 3.2. pH Optimisation of Assay

One likely reason for the relatively poor difference in rates between the enzymatic and uncatalysed reaction from the originally reported assay method [10] was the relatively high pH in which they were carried out. Thus, the effect of pH on the hydrolysis of **1** was first investigated. Silα, fused with the trigger factor at the *N*-terminal and a Strep-tag II affinity tag at the *C*-terminal (henceforth referred to as TF-Silα-Strep) was used as the model enzyme. The data from the enzymatic reactions were then compared to non-enzymatic hydrolysis, where the enzyme had been omitted from the reaction mixtures. As expected, it was found that the rate of non-enzymatic hydrolysis increased with pH (Figure 1A). A similar trend was observed with the enzymatic reaction, but once the net rate was calculated, the enzyme was found to exhibit a good activity between 6.5–8.5 (Figure 1B), with an optimal pH of 8.5. This value is consistent with the optimal pH range for serine proteases such as chymotrypsin and trypsin, as well as the alkaline proteases [9,19,20].

### 3.3. Substrate Screening

The hydrolysis of the new substrates **4**–**10** was subsequently tested in a similar manner at pH 8.5. However, during these experiments, it was found that the azobenzene-derived substrate **10** and methyl substituted cyanophenol substrates **8** and **9** were essentially insoluble in the reaction mixture, even though it already contained 10% *v*/*v* 1,4-dioxane as a co-solvent. DMSO was also tested as an alternative biocompatible solvent (at the same concentration), but was found to be ineffective, and these substrates were therefore excluded from further investigation. Of the remaining substrates, the reaction conversions were quantified by UV–VIS spectrophotometry (Appendix A), and their corresponding initial rates were calculated (Figure 2, Appendix A).

In the assays employing the 50 μM substrate (the same concentration as used in previous work), the ratio of enzymatic to non-enzymatic rates (Figure 2A) showed that **4** provided the greatest differentiation between the enzymatic reaction and the uncatalysed hydrolysis. Here, a ratio of 6.2 between the two rates was achieved, although in both (enzymatic and uncatalysed) the absolute rates were lower compared with the parent substrate **1**. The slower rates with the 2-methyl substituted (i.e., *ortho* relative to the scissile Si-O bond) **4** were likely due to the mildly electron-donating effect of this substituent, as well as steric blocking. Significantly, the uncatalysed reaction appeared to be retarded to a greater degree than the enzymatic reaction, although the reason was unclear. The 3-methyl substrate **5** provided uniformly lower rates for the enzymatic and non-enzymatic reactions compared with the parent **1**. This result is consistent with the fact that the methyl group prevents the neighbouring nitro group from being coplanar with the ring, thereby making the nitrophenoxide a poorer leaving group. 

In contrast, the methoxy-substituted **6** provided the highest rates for both reactions, so clearly the inductive electron-withdrawing effects were more important than any mesomeric stabilisation of the putative-leaving group. The 4-cyanophenoxy substrate **7** was anticipated to provide much lower rates of reaction compared with **1,** given the cyano group’s less negative Hammett σ constant (−0.56) compared with the nitro group (−0.71) [21], and this was indeed observed in the experimental results. Although the addition of the enzyme did provide a higher rate of hydrolysis for substrate **7** compared with when the enzyme was omitted, the difference was the smallest among the tested substrates, suggesting that **7** was less well accepted by the enzyme.

To assess the effect of the substrate concentration, which may affect enzyme occupancy (see below), the assay was then carried out with 100 µM of substrate (Figure 2B, Appendix A and Appendix A). As expected, the absolute rates were found to be higher in all cases due to the increase in substrate concentration. However, it was also found that the ratio of enzymatic to uncatalysed reactions were improved for substrates **4** and **5**. In the case of **4**, a dramatic increase to a ratio was observed with the enzymatic reaction being nearly 12-fold greater than the uncatalysed reaction. These results suggest that the substrate binding to the enzyme is relatively weak and at the lower substrate concentration, enzyme occupancy (and hence enzyme-catalysed turnover) was suboptimal. 

Overall, the ratios of enzymatic to background reaction rates for substrates **5**, **6**, and **7** were inferior to the original substrate **1**. Only substrate **4** produced an improved result that could be translated to a superior signal-to-noise ratio in the subsequent Michaelis–Menten kinetics measurements (see below).

### 3.4. Kinetic Analysis of TF-Silα-Strep on Silyl Ether Hydrolysis

To demonstrate the utility of the new substrates for quantitative enzyme kinetics analyses, the molecules **4** and **5** that displayed the two best rate ratios were then applied in enzyme assays to determine their Michaelis–Menten kinetic parameters with respect to TF-Silα-Strep (using the net rates of reaction, as previously reported). For consistency, the kinetic parameters for the previously reported substrates **1**–**3** were also determined. The results obtained from the Michaelis–Menten plots (Table 1, Appendix A) showed that all of the substrates had *K_M_* in the μM range, which were consistent with the results previously reported for **1** [10]. The 2-methyl substrate **4** exhibited a *K_M_* of 72.5 μM, which was substantially higher than the other substrates (i.e., weaker binding or fewer productive binding events). This result is consistent with the substrate screening experiments above, whereby the use of 100 μM of substrate (i.e., above the *K_M_*) provided an improvement in the ratio of catalysed to uncatalysed hydrolysis.

In comparing the *k_cat_* of the TBDMS-bearing substrates, the unsubstituted **1** provided the highest rate constant, which was unsurprising as the introduction of the methyl groups in **4** and **5** were intended to reduce their susceptibility to hydrolysis. In comparing **1**–**3**, increasing steric bulk of the silyl groups reduced *k_cat_* due to the reduction in accessibility of the Si–O bond. This steric bulk in **2** and **3** had a much greater effect than the methylations in **4** and **5**.

## 4. Conclusions

In summary, this study reports the development of chromogenic substrates that are applicable for the spectrophotometric quantification of biocatalytic Si-O bond hydrolysis. A series of molecules bearing a sessile Si-O bond attached to a chromophore were synthesised, characterised, and, where possible, their rates of hydrolysis were measured under conditions that are suitable for the enzymatic assay. Here, the ratio of the initial rates of the enzymatic and non-enzymatic reactions was used as the key criterion for determining the optimal substrate and reaction conditions. 

It was found that for the model silicatein enzyme, *tert*-butyldimethyl(2-methyl-4-nitrophenoxy)silane (**4**) displayed the best rate ratio of enzyme-catalysed to uncatalysed reactions, albeit with lower absolute rates. This finding underscores the potential of employing tailored spectrophotometric substrates to provide a better signal-to-noise ratio for improved accuracy. However, the subsequent Michaelis−Menten kinetics analysis demonstrated that care should be taken in the selection of substrates so they are matched to the binding affinity of the enzyme of interest. Nevertheless, from a practical perspective the use of lower quantities of substrate to conserve reagents may still find utility as a qualitative assay.

Potential avenues for future work include an exploration of substrate diversity to include alternative chromophores and the investigation of fluorogenic assays for broader analytical applications and further improved sensitivity. In terms of wider applications, the molecules reported here could also be used as part of an assay for the detection of fluoride ions [18].

## Data Availability

Numerical data are available from the authors upon request.

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
