# Peer review of "Development of Improved Spectrophotometric Assays for Biocatalytic Silyl Ether Hydrolysis"

_biomolecules, 2024, doi:10.3390/biom14040492_

Round 1
Reviewer 1 Report
Comments and Suggestions for Authors
Lu et al explore the use of different phenoxy silyl ester derivatives as research tools for characterising hydrolytic enzymes. The manuscript is very well-presented and I commend the authors for the clarity of their writing.
The methods are described at an appropriate level of detail, building upon previous work in many cases. Experiments are carefully performed with appropriate controls.
The results show a pH optimum for catalysis, and identify one substrate with attractive characteristics (low background hydrolysis, and so better signal-to-noise). I have only a couple of very minor comments here.
1. Regarding the kinetics, it would be helpful for the authors to state that they performed continuous assays; the text at the moment implies that only a single timepoint was samples (30 mins). Could they also please state why they chose to select a single timepoint rather than using the more accurate fitting of the linear portion of the curve? For example, for Fig. S2B, just taking the signal at 30 mins does not reflect the actual rate of change of the reaction once it enters the measurable range.
2. Just a note for the future – it is always preferable for control experiments to use boiled protein rather than simply to omit protein. Did the authors ever try this?
Reviewer 2 Report
Comments and Suggestions for Authors
This is a very succinct paper demonstrating the development of 10 new chromophores for detecting cleavage of the Si-O bond. I believe this paper is all written and appropriate for publication.
One minor comment relates to Figure 1 - the review suggests to use the same analysis that is demonstrated in later plots, comparing the ratio of uncatalyzed vs catalyzed silane reduction. This will streamline the results.
Reviewer 3 Report
Comments and Suggestions for Authors
The authors describe an exciting idea for improving an enzymatic assay to quantify Si-O hydrolysis. However, some adjustments must be made for this article to be accepted.
-
Lines 34-37 - font size and type appear to be different from the text;
-
The English need to be reassessed, for example:
Lines 148-151- …"substrates were designed that were intended"... the sentence needs to be rewritten.
Line 159 "visual spectrum should be exchanged for "visible region of the spectrum"
However several other parts of the text should be reassessed.
-
The authors should withdraw compounds 8 and 9 of the discussion and the paper or perform the test using a cosolvent (enzyme-friendly cosolvent). I strongly suggest the last one.
-
Lines 202-204 the authors say "The slower rates with the 2-methyl substituted (i.e. ortho relative to the scissile Si-O bond) 4 were likely due to the mildly electron-donating effect of this substituent, as well as steric blocking. " But the rate for compound 5, for example, is slower. How do you explain that? The author should clarify that is only the uncatalyzed reaction that is affected. As it is the sentence is confusing.
-
Line 220, How does the concentration of 100 µM compare to the other published assays? Is this acceptable?
-
In the conclusions, the authors say that compound 4 displayed the best rate ratio of catalyzed/uncatalyzed. Still, the Km for this substrate was substantially higher than the others and this should be an issue to be overpassed.
After all this observation, this reviewer thinks that the work is not done for publication.
Comments on the Quality of English LanguageThe quality of the text in general strongly needs to be improved.
Round 2
Reviewer 3 Report
Comments and Suggestions for Authors
The authors answered all the comments.